# Synthesis, Cytotoxic, and Computational Screening of Some Novel Indole–1,2,4-Triazole-Based *S*-Alkylated *N*-Aryl Acetamides

**DOI:** 10.3390/biomedicines11113078

**Published:** 2023-11-16

**Authors:** Ameer Fawad Zahoor, Sadaf Saeed, Azhar Rasul, Razia Noreen, Ali Irfan, Sajjad Ahmad, Shah Faisal, Sami A. Al-Hussain, Muhammad Athar Saeed, Muhammed Tilahun Muhammed, Zeinab A. Muhammad, Magdi E. A. Zaki

**Affiliations:** 1Department of Chemistry, Government College University Faisalabad, Faisalabad 38000, Pakistan; fawad.zahoor@gcuf.edu.pk (A.F.Z.); raialiirfan@gmail.com (A.I.);; 2Department of Zoology, Government College University Faisalabad, Faisalabad 38000, Pakistan; 3Department of Biochemistry, Government College University Faisalabad, Faisalabad 38000, Pakistan; 4Department of Health and Biological Sciences, Abasyn University, Peshawar 25000, Pakistan; 5Gilbert and Rose-Marie Chagoury School of Medicine, Lebanese American University, Beirut P.O. Box 36, Lebanon; 6Department of Natural Sciences, Lebanese American University, Beirut P.O. Box 36, Lebanon; 7Department of Chemistry, Islamia College University Peshawar, Peshawar 25120, Pakistan; 8Department of Chemistry, College of Science, Imam Mohammad Ibn Saud Islamic University (IMSIU), Riyadh 11623, Saudi Arabia; 9Department of Pharmaceutical Chemistry, Faculty of Pharmacy, Suleyman Demirel University, Isparta 32000, Türkiye; 10Department of Pharmaceutical Chemistry, National Organization for Drug Control and Research (NODCAR), Giza 12311, Egypt

**Keywords:** indole–triazole, anticancer, Hep-G2 cancer cell line, hemolysis, thrombolysis, SAR, in silico profiling, ADMET studies, DFT studies

## Abstract

Molecular hybridization has emerged as the prime and most significant approach for the development of novel anticancer chemotherapeutic agents for combating cancer. In this pursuit, a novel series of indole–1,2,4-triazol-based *N*-phenyl acetamide structural motifs **8a**–**f** were synthesized and screened against the in vitro hepatocellular cancer Hep-G2 cell line. The MTT assay was applied to determine the anti-proliferative potential of novel indole–triazole compounds **8a**–**f**, which displayed cytotoxicity potential as cell viabilities at 100 µg/mL concentration, by using ellipticine and doxorubicin as standard reference drugs. The remarkable prominent bioactive structural hybrids **8a**, **8c,** and **8f** demonstrated good-to-excellent anti-Hep-G2 cancer chemotherapeutic potential, with a cell viability of (11.72 ± 0.53), (18.92 ± 1.48), and (12.93 ± 0.55), respectively. The excellent cytotoxicity efficacy against the liver cancer cell line Hep-G2 was displayed by the 3,4-dichloro moiety containing indole–triazole scaffold **8b,** which had the lowest cell viability (10.99 ± 0.59) compared with the standard drug ellipticine (cell viability = 11.5 ± 0.55) but displayed comparable potency in comparison with the standard drug doxorubicin (cell viability = 10.8 ± 0.41). The structure–activity relationship (SAR) of indole–triazoles **8a**–**f** revealed that the 3,4-dichlorophenyl-based indole–triazole structural hybrid **8b** displayed excellent anti-Hep-G2 cancer chemotherapeutic efficacy. The in silico approaches such as molecular docking scores, molecular dynamic simulation stability data, DFT, ADMET studies, and in vitro pharmacological profile clearly indicated that indole–triazole scaffold **8b** could be the lead anti-Hep-G2 liver cancer therapeutic agent and a promising anti-Hep-G2 drug candidate for further clinical evaluations.

## 1. Introduction

Cancer is a diverse group of dreadful diseases characterized by uncontrolled cell growth and is a major health problem worldwide [1,2]. In 2018, the World Health Organization (WHO) reported that cancer is the leading cause of death, and the number of cancer cases will increase to 22 million by 2030 [3]. The fatality rate as a result of leukemia, colon cancer, and liver cancer is found to be reasonably higher [4,5]. Particularly, hepatocellular carcinoma (HCC) is the most common form of liver cancer. It is estimated to account for 75% of all primary liver cancers; it is the sixth most prevalent fatal malignancy, with almost one million deaths annually; and it is the second main cause of all deaths relevant to cancer [6,7,8]. The high mortality rate associated with HCC can be largely attributed to its widespread prevalence and the inadequacy of conventional chemotherapy as an effective treatment [9]. FDA-approved drugs such as Sorafenib and Regorafenib were used as multi-kinase inhibitors for the treatment of HCC; however, this treatment can only improve patients’ median survival for about 3 to 10 months [10,11]. To overcome these challenges, the development of new and efficient chemotherapeutics is necessitated in medicinal chemistry, despite the availability of several drugs to treat different cancers [12,13,14,15,16].

Nitrogen-containing heterocyclic compounds have gained prominence in the synthetic and pharmaceutical industries because of their diverse biological actions [17]. For example, these actions include anti-microbial [18,19], anticancer [20,21], anti-malarial [22], anti-HIV [23], anti-diabetic [24], and anti-tubercular activities [25]. Among the nitrogen-containing heterocycles, the indole scaffolds possess remarkable anti-tuberculosis [26], anti-inflammatory [27], and anticancer activities [28]. The intriguing indole moiety is well-known for its anticancer activities and occurs widely in numerous natural products such as vinblastine and vincristine, which can be used to treat breast cancer, testicular cancer, ovarian cancer, and neck and head cancer [29,30,31]. There is significant evidence that indole can be linked to a variety of other anticancer compounds, such as 1,2,4-triazole, to enhance their anticancer activities. In addition, 1,2,4-triazole exhibits pharmacological activity and is present in various drugs, including letrozole as an anti-breast cancer agent, ribavirin as an anti-viral agent, fluconazole as an anti-fungal agent, etc. Indole is also an integral part of many clinical drugs such as semaxanib, indibulin, rizatriptan, etc., as displayed in Figure 1 [32,33,34,35].

### Molecular Fragmentation Hybridization Approach for Rationale Design 

Recently, new molecules of indole with structural modification have emerged possessing potent activity against carcinoma, and our group reported different heterocyclic S-linked *N*-phenyl acetamide scaffolds as anti-HepG-2 cancer agents, which are the basis of this research work (Figure 2) [36,37,38].

Inspired by these aforementioned findings and our investigation into the preparation of new heterocycles as anticancer agents, we synthesized a series of novel indole–1,2,4-triazole-based *S*-alkylated tethered *N*-arylacetamides and evaluated their anticancer activities against a human liver tumor cell line (Hep-G2). Moreover, in silico approaches like molecular docking scores, molecular dynamic simulation stability data, DFT studies, and the ADMET pharmacological profile further supported the anti-Hep-G2 potential of synthesized compounds.

## 2. Materials and Methods

### 2.1. Materials for the Synthesis of Indole–1,2,4-Triazole Structural Hybrids

All analytical-grade chemicals were available commercially from Alfa Aesar (Ward Hill, MA, USA), Sigma Aldrich, and Merck (Burlington, MA, USA), and used without further purification. Solvents were purified through distillation. TLC was used to monitor the initial purity of the synthesized compounds and to detect the progress of the reaction. The melting points of synthesized derivatives were taken on Gallenkamp equipment. For the characterization of target compounds, ^1^H-NMR spectra (δ = ppm) at 400 MHz, Bruker DPX-400, and ^13^C-NMR spectra (δ = ppm) at 100 MHz were obtained with a Bruker (Bruker, Zurich, Switzerland), spectrophotometer.

### 2.2. Synthetic Strategies and Procedures

#### 2.2.1. Synthesis of Ethyl 2-(1H-indol-3-yl) Acetate 2

A solution of 2-(1*H*-indol-3-yl) acetic acid **1** (3 g, 17.1 mmol) in ethanol (20 mL) was cooled to 0 °C, to which a catalytic amount of H_2_SO_4_ was added and refluxed for 6 h. After completion of the reaction, monitored by TLC, the reaction mixture was basified with a saturated solution of sodium carbonate. The organic phase was collected using ethyl acetate (50 mL × 3), dried by Na_2_SO_4_, and concentrated by rotary evaporation. Crude solids were exposed to column chromatography using silica gel to furnish pure acetate **2** (Figure 1) [38].

#### 2.2.2. Synthesis of 2-(1*H*-indol-3-yl) Acetohydrazide **3**

To a solution of ethyl 2-(1*H*-indol-3-yl) acetate **2** (0.2 g, 0.98 mmol) in 30 mL ethanol, hydrazine hydrate (0.19 g, 3.93 mmol) was added slowly at 0 °C and stirred overnight at r.t. After completion of the reaction (indicated by TLC), ethanol was evaporated, and pure 2-(1*H*-indol-3-yl) acetohydrazide **3** was achieved [38] (Figure 1).

#### 2.2.3. Synthesis of 5-((1H-indol-3-yl)methyl)-4-(3,4-dichlorophenyl)-4H-1,2,4-triazole-3-thiol **6**

A mixture of 3,4-dichlorophenyl isothiocyanate **4** (0.21 g, 1.05 mmol) and hydrazide **3** (0.2 g, 1.05 mmol) in 20 mL of ethanol was refluxed for 2 h. The obtained intermediate **5,** 3, 4–dichlorophenyl thiosemicarbazide precipitate was filtered off after cooling, washed with ethanol, and dried. The resultant intermediate **5** was added to a KOH (0.08 g, 1.58 mmol) solution in water (5 mL) and allowed to reflux for 4 h. After cooling, diluted HCl was added to the solution to make it acidic, and the formed precipitate was filtered out. Recrystallization of precipitate from ethanol provided the title triazole-3-thione **6** as a white solid [39,40], as depicted in Figure 1.

#### 2.2.4. Synthetic Procedure to Synthesize Indole–1,2,4-Triazole Structural Hybrids **8a**–**f**

To a solution of triazole-3-thione **6** (0.2 g, 0.53 mmol) in dichloromethane, pyridine (0.08 g, 1.06 mmol) was added. After complete addition, the resulting reaction mixture was stirred for 15 min. 2-bromo-*N*-arylacetamide **7a**–**f** (0.85 mmol) was then added to the reaction mixture and allowed to stir at 25 °C for 24 to 48 h, as in Figure 1. Then, *n*-hexane was added to obtain the precipitates of product **8a**–**f**, which were purified by recrystallization from ethanol [41].

#### 2.2.5. Spectral Data of the Synthesized Compounds **8a**–**f**

All the characterization data such as ^1^H and ^13^C NMR, molecular weight, melting points, and percentage yields of synthesized compounds are given as the Appendix A.

### 2.3. In Vitro Anticancer Activity

The cytotoxic activity of indole hybrids was investigated against liver cancer cells (Hep-G2). Cell viability was determined using the MTT assay using elipticine [42] and doxorubicin [43]. The corresponding cancer cells were grown at 37 °C in Dulbecco’s Modified Eagle Medium enriched by streptomycin (100 µg/mL) and 10% fetal bovine serum along with penicillin (100 units/mL). A 0.05% concentration of DMSO (dimethyl sulfoxide) was used to dissolve indole derivatives and was then subjected to cells. Next, 500 µg/mL MTT reagent was added, and the cells were incubated for 4 h. DMSO (150 µL) was used to dissolve the Formazan crystals, and absorbance was recorded at 570 nm [44].

### 2.4. Molecular Docking of Indole–1,2,4-Triazoles

In computational studies to identify the probable targets for these compounds, first the structures of these indole–triazole hybrid compounds were prepared using ChemDraw Professional, and then the SMILES of these compounds were uploaded to the “Swiss-Target Prediction” and “ACID Reverse Docking” online servers, where the initial probable cancer macromolecular targets were shortlisted [45,46,47]. After the identification of the possible macromolecular cancer targets for these compounds, to further evaluate the binding conformations and affinities of these compounds with these macromolecular targets, we retrieved the PDB structures of the enzyme targets PKC-θ, AKT1, PI3K, and VEGFR2 from the RCSB PDB Bank with PDB IDs 1XJD 4EKL, 5DXU, and 3VHE.49 [48,49,50,51]. The conventional Induced-Fit molecular docking was performed using the Molecular Operating Environment (MOE) 2015.10 software [52], and for visualization of the protein–ligand complexes, the Biovia DS (v2017) software was used.

### 2.5. ADMET and Drug-Likeness Studies of Indole–1,2,4-Triazoles

The prediction of the pharmacokinetics, medicinal, and toxicological studies in the ADMETlab 2.0 online server was utilized in this study [53,54].

### 2.6. MD Simulation of the Most Bioactive Indole–1,2,4-triazoles

Deciphering the structure dynamics of a biomolecule in the presence of a ligand molecule is a key feature in drug discovery as it unfolds many vital aspects to guide lead compound optimization [55]. The molecular dynamics simulation study was conducted using AMBER v20 software [56]. The preprocessing of the proteins and compounds was conducted using the AMBER antechamber program [57]. The topology files of proteins were defined by the FF14Sb force field, while those of compounds were generated using the AMBER general force field (GAFF) [58,59]. The energy minimization process for the complexes was completed using 1000 rounds of steepest descent and 1500 rounds of the conjugate gradient algorithm. The complexes were then accommodated in the TIP3 water box, keeping the padding distance of 12 Å. The systems were neutralized by adding solvents. The number of sodium ions was 12, 10, and 14 for the AKT1-8a complex, the AKT1-8b complex, and the AKT1-8f complex, respectively. The docked complexes were equilibrated at 300 K in the NVT ensemble for 500 ps. The production run was carried out using 100 ns, considering the time size set to 2 fs. During the production run, temperature control was accomplished using the Langevin algorithm, while hydrogen bonds were constrained using the SHAKE method [60,61]. The long-range electrostatic interactions were calculated by the particle mesh Ewald method. The trajectory analysis comprises root-mean-square deviation (RMSD) and root-mean-square fluctuation (RMSF), both of which were carried out considering carbon alpha atoms [62,63]. The CPPTRAJ module of Amber was used for the structure analysis discussed above. XMGRACE v5.1 was employed for plotting purposes [64,65].

### 2.7. MMPBSA/MMGBSA Binding Free Energies Estimation

The relative binding free energies of the complexes were estimated using the MMPBSA.py script of AMBER v20 [66]. The net binding energy of docked complexes and the contribution of van der Waals and electrostatic energies were predicted using the MMGBSA method [67]. The method estimates the free energy using the following formula:∆G binding = ∆G complex − [∆G receptor + ∆G ligand]

For the energy calculation, a total of 1000 frames were selected from trajectories.

### 2.8. DFT Studies

DFT (density functional theory) computations were performed using the Gaussian 09 program [68]. First, compounds **8a**, **8b**, and **8f** were optimized through the DFT method with B3LYP in the ground state [69]. The 6-311G-++-d-p basis set was utilized in the computation. Thereafter, the energy computations of compounds **8a**, **8b**, and **8f** were carried out by preserving the optimization DFT setups. In the final step, GaussView 5.0 was used to analyze the DFT computation results [70]. For this end, molecular electrostatic potential (MEP), highest occupied molecular orbital (HOMO), lowest unoccupied molecular orbital (LUMO) analysis, and calculation of the related energies were performed [37].

## 3. Results and Discussion

### 3.1. Chemistry

The synthesis of novel indole-linked 1,2,4-triazole derivatives **8a–f** was efficiently performed according to the synthetic strategies depicted in Figure 1. Initially, esterification of commercially available indole-3-acetic acid 1 with ethanol utilizing H_2_SO_4_ as a catalyst led to the formation of the corresponding ester 2 in a 79% yield. Subsequently, hy-drazinolysis of ester 2 with hydrazine hydrate in ethanol afforded the 2-(1*H*-indol-3-yl)acetohydrazide 3 in a 96% yield [38]. The 2-(1*H*-indol-3-yl)acetohydrazide 3 upon treatment with 3,4-dichlorophenyl isothiocyanate 4 yielded intermediate 5, which under reflux in KOH furnished the desired indole-linked 1,2,4-triazole-3-thione 6 in a 75% yield [39,40]. Finally, the alkylation of the resultant 1,2,4-triazole-3-thione 6 with various 2-bromo-*N*-arylacetamides **7a**–**f** in DCM and pyridine at room temperature accomplished the target products **8a**–**f** in a 69–82% yield (Figure 1) [41].

### 3.2. Spectral Interpretation of Compound ***8b***

Spectral confirmation of synthesized derivative **8b** was conducted by ^1^H-NMR and ^13^C-NMR spectroscopic techniques. The ^1^H-NMR spectrum of **8b** depicted a singlet peak for the indole NH proton at δ 10.84 ppm, and another singlet appeared at δ 10.41 ppm for the anilide NH proton. At δ 4.05 ppm and δ 4.13 ppm, signals appeared for methylene linkers. Multiplet signals appeared at δ 6.76–7.67 for aryl and indole ring protons. The ^13^C-NMR spectrum of **8b** also supported the carbon skeleton, and exhibited a signal at δ 165.77 ppm for the carbonyl group. The presence of two methylene linkers was verified by two upfield singlets at δ 21.71 and δ 37.45. Two downfield signals that confirmed the formation of the 1,2,4-triazole ring appeared at δ 155.06 and δ 149.56. The 3,4-dichlorophenyl moiety linked to the triazole ring displayed signals at δ 132.90, δ 132.81 for C-Cl, and δ 129.48 for C-N, while the other three signals appeared at δ 131.34, δ 126.63, and δ 123.66. Signals depicting the formation of indole rings were observed at δ 137.76, δ 127.75, δ 127.19, δ 121.21, δ 118.53, δ 118.21, δ 111.45, and δ 107.86. The carbons of the 4-chlorophenyl ring attached to acetamide indicated signals at δ 131.94 for C-Cl, and the remaining carbons showed signals at δ 136.06, δ 128.81, and δ 120.74.

All other synthesized derivatives in the series (**8a**–**f**) were characterized structurally using a similar approach.

### 3.3. Anti-Hep-G2 Activity

In this study, the cytotoxic activity of tested indole-based triazole hybrids was evaluated against a human liver tumor cell line (Hep-G2) in vitro through an MTT assay using ellipticine [42] and doxorubicin [43] as reference standard drugs. Remarkable results were obtained by applying one dose of the targeted compounds, and are presented in Table 1. It was evident from the obtained cell viability values that most of the synthesized compounds displayed a good-to-high cytotoxic effect against the tested cell line. Among all derivatives, compound **8b**, bearing the chloro group in the *para* position on the anilide ring, proved to be the most potent cytotoxic agent by displaying the lowest cell viability value, i.e., 10.99 ± 0.59% at 100 µg/mL concentration. The introduction of the fluoro group at the *ortho* position of *N*-arylacetamide, as in compound **8f** (cell viability = 12.93 ± 0.55%), with an IC_50_ value of 55.40 μg/mL, also possessed promising cytotoxic potential. Compound **8a** (cell viability = 11.72 ± 0.53%), bearing two methyl groups at the *meta* and *para* positions of the anilide ring, also showed good inhibition potential. However, moderate cytotoxic activity results were obtained in the cases of compound **8c** possessing the methoxy groups on the anilide ring at the *ortho* and *para* positions (cell viability = 18.92 ± 1.48%) and compound **8d** having ortho-methoxy (cell viability = 38.92 ± 6.75%). Furthermore, different concentrations of the tested compound **8f** were applied to cancerous cells to check the dose-dependent cytotoxic nature of these indole derivatives. The cell viability of 7.98% was observed at a concentration of 200 µg/mL, which considerably highlighted the significance of these derivatives.

### 3.4. Structure–Activity Relationship (SAR)

The structure–activity relationships (SAR) study revealed the significance of anilides, which were used to increase the lipophilic nature of indole derivatives, as displayed in graphical form in Figure 3.

It was observed that the unsubstituted anilide ring does not impart any anticancer therapeutic efficacy to compound **8e** by displaying the highest cell viability value of 123.21 ± 2.16%. The 4-chloro-substituted anilide ring of indole–triazole scaffold **8b** remarkably enhanced the cytotoxicity therapeutic potential by exhibiting the lowest (10.99 ± 0.59%) cell viability compared with the standard drug ellipticine (cell viability = 11.5 ± 0.55%), but displayed comparable potency in comparison with the standard drug doxorubicin (cell viability = 10.8 ± 0.41%), as depicted in Figure 3. The 3,4-dimethyl moiety containing indole–triazole structural motif **8a** displayed good anti-Hep-G2 potential, with a cell viability of 11.72 ± 0.53%. The highest electronegative 2-fluoro-substituted indole–triazole compound **8f** displayed better and comparable cytotoxic chemotherapeutic potential, with 12.93 ± 0.55% cell viability and 55.40 µg/mL IC_50_ values, as depicted in Figure 4. The structure–activity relationship revealed that the presence of electronegative chemical entities such as fluorine and chlorine on the phenyl ring at the second and fourth positions of *S*-linked indole–triazoles **8f** and **8b** significantly enhanced the anti-Hep-G2 efficacy. The methyl moiety at the third and fourth positions of the phenyl ring also enhanced the anticancer potential of indole–triazole derivatives, as displayed by compound **8a**.

### 3.5. In Silico Molecular Docking Studies

We used in silico computational probing approaches to find and investigate the likely molecular targets involved in malignancies for the indole–triazole hybrids, which earlier demonstrated good activity against the examined liver cancer cell line. First, the indole–triazole scaffolds **8a**, **8b**, and **8f**, which showed excellent-to-good anticancer activities, were screened against a repertoire of different classes of enzymes involved in various biological activities using the reverse, or sometimes also called the inverse molecular docking/virtual screening approaches. To determine the probable binding cavities of a group of therapeutically significant macromolecular targets involved in cancer for these compounds, the reverse docking programs “Swiss-Target Prediction and “ACID Reverse Docking” were used [46,47]. This reverse docking approach showed that these compounds show good binding with the kinase family of enzymes, which are important biological enzymes and control different cellular signaling and proliferation pathways. As a result of several kinase enzymes being overexpressed in many cancers and contributing to nearly all stages of cancer, various anticancer medications target these kinases to treat different malignancies [71].

By using the aforementioned two inverse docking servers, we were able to identify several kinase enzymes that are reported in the literature to have a significant role in cancer. The analysis of the inverse docking studies revealed that PKC-θ, AKT1, PI3K, and VEGFR2 are the most probable macromolecular targets for these compounds. These enzymes also happen to be a hotspot for the development of anticancer drugs [46,47,71], and numerous compounds with the indole core moiety have been discovered to have inhibited these macromolecular cancer targets [47,71,72,73,74,75,76].

We further carried out the traditional Induced-Fit molecular docking investigations to determine the binding mechanisms and affinities of these indole-based compounds with the identified macromolecular cancer targets. These molecular docking investigations were carried out using all the identified potential enzyme targets of the synthesized indole–triazole compounds that had shown good activity in the in vitro studies involving the Hep-G2 liver cancer cell line.

The results of the molecular docking studies revealed that out of all the shortlisted possible kinase enzyme targets, these indole–triazole hybrids showed good binding affinities with the AKT1 kinase enzyme. The analysis of their binding affinities showed that the indole–triazole hybrids **8a**, **8b**, and **8f** were able to show higher affinities than the control drug Ipatasertib of the AKT1 enzyme in the molecular docking studies.

The Ipatasertib inhibitor of the AKT1 enzyme was able to show a binding affinity of −8.25 Kcal/mol, while the newly synthesized compounds **8a**, **8b**, and **8f** were able to bind with the AKT1 with binding affinities of −8.31 Kcal/mol, −8.33 Kcal/mol, and −8.60 Kcal/mol, respectively.

The conformational binding and interaction analysis of these indole–triazole hybrids showed that these compounds show robust interactions of multiple types with the active site of the AKT1 target enzyme. It was seen in the AKT1+**8a** ligand–protein complex that this compound interacted via conventional hydrogen bonds and carbon–hydrogen-type H-bonds by engaging the GLY159, LYS179, and PHE161 active site receptor residues with its triazole and acetamide moieties. The indole core of the **8a** showed Pi-alkyl interactions with the VAL164 and MET281, while the phenyl moiety present on the triazole of this compound, as well as the phenyl group attached to the acetamide functionality, was also able to make multiple Pi-Anion, Pi-Cation, Pi-Pi T-shaped, and Pi-Alkyl interactions with the HIS194, PHE161, LYS179, GLU278, ASP292, and PHE442 receptor residues of the AKT1 multiple times. Several van der Waals were also observed in the AKT1+**8a** complex. The AKT1+**8a** 2-dimensional and 3-dimensional representations are presented in Figure 5.

Similarly, the conformational and interaction analyses of the other two complexes, **AKT1+8b** and **AKT1+8f**, revealed that these compounds also possess the same type of conformation and robust interactions with the target enzyme active site. The graphical 2-dimensional representations of both of these complexes can be seen in Figure 6, where the indole core scaffold, the triazole, -arylacetamide, and the phenyl attached to the triazole moiety of these compounds make multiple types of interactions. The strong hydrogen bond interactions, as well as other types of hydrophobic interactions along with Pi-Sulfur and van der Waals types of interactions, were also present in these enzyme complexes.

The chemical structures of **8a**, **8b**, and **8f,** along with their binding affinities with the target AKT1 enzyme, are presented in Table 2, while the binding affinities of these compounds with the other studied enzyme targets (PKC-θ, PI3K, and VEGFR2) are available in the Appendix A.

### 3.6. In Silico ADMET Studies

These substances were also assessed for pharmacokinetics and drug-likeness using the in silico predictive techniques. These investigations showed that these compounds possess the optimal molecular weights, TPSA scores, number of hydrogen bond acceptors and donors, and number of rings (nHA, nHB, and nRing) that are required for a bioactive compound. According to ADMETlab2.0 predictions, these substances demonstrated reduced permeability in MDCK (Madin–Darby Canine Kidney) cells and optimal human intestine absorption (HIA) in the in silico predictive models. The Pfizer drug rule was fully complied with, and all of the compounds displayed favorable medicinal chemistry profiles.

These substances are AMES toxic, according to the results of the toxicity studies, and should not be taken during pregnancy; however, these substances agreed to the acute toxicity rule and had a low oral toxicity profile in rats and were also less carcinogenic. According to the predicted metabolism studies, these substances are substrates for the metabolic transformation enzyme CYP3A4. According to the excretion prediction profiles for these chemicals, the host excretory system clears them at a moderate rate (5–15 mL/min/kg). Table 3 provides data on the top three compounds’ ADMET and drug-likeness profiles, as predicted by ADMETlab2.0.

### 3.7. In Silico Molecular Dynamic Simulations

#### 3.7.1. Dynamic Structure Analysis

The molecular dynamic simulation is a powerful technique to study the biomolecular structure–function relationship and, in particular, yields important underpinnings for ligand binding and interactions. The simulation analyses were conducted on a time scale of 100 ns (Figure 7). All three complexes were found to behave similarly, with minor structural changes. The AKT1-**8a** complex was found to show a lower RMSD compared to the AKT1-**8b** complex and the AKT1-**8f** complexes. The average RMSD of the AKT1-**8a** complex, AKT1-**8b** complex, and AKT1-**8f** complex is 0.8 Å, 1.0 Å, and 1.4 Å, respectively. The initial deviations of complexes were due to small structure adjustments by the protein to deeply accommodate the ligands. However, towards the end, the complexes gained stable dynamics. The RMSD is supported by the RMSF plots, which depict stable residue fluctuations during simulation time. The average RMSF of the AKT1-**8a** complex, AKT1-**8b** complex, and AKT1-**8f** complex is 1.0 Å, 1.2 Å, and 1.3 Å, respectively.

#### 3.7.2. Binding Free Energies

The complexes were noticed to show significant stability, with net MMGBSA binding energy values of −65.86 Kcal/mol (AKT1-8a complex), −71.38 Kcal/mol (AKT1-8b complex), and −58.9 Kcal/mol (AKT1-8f complex). The values demonstrate that the docked complexes favor strong intermolecular binding conformations and short-distance bonding. Both van der Waals and electrostatic energies favor intermolecular complex formation and play a high role in overall complex stability. The solvation energy was seen as contributing less to the overall net energy. The non-polar solvation energy, especially, was unfavorable. The overall energy terms for complexes are tabulated in Table 4.

### 3.8. In Silico DFT Studies

#### 3.8.1. Molecular Electrostatic Potential Appraisal

The DFT study yields various electrochemical properties for compounds. The molecular electrostatic potential (MEP) is among these properties that are used to estimate the reactivity of a compound [77]. In the MEP distribution maps, red and yellow parts represent electrophilic reactivity, whereas blue represents nucleophilic reactivity. For the three compounds, predominantly yellow and red parts were observed around the oxygen of the carbonyl in the acetamide group (Figure 8). Similarly, a predominantly yellow part was observed on the phenyl of the indole ring. Hence, the electrophilic reactivity of the compounds might result from these vicinities. On the other hand, predominantly blue parts were observed around the triazole ring and the phenyl of the chlorophenyl ring (Figure 8). Therefore, the nucleophilic reactivity of the compounds might result from these vicinities.

#### 3.8.2. HOMO-LUMO Evaluations

The DFT study yields HOMO-LUMO energies. Based on these values, the related energies were calculated using the appropriate formulas. The HOMO-LUMO energies are used to evaluate the electrical properties and chemical affinities of compounds. The electron donor property is represented by the HOMO energy, whereas the electron acceptor property is represented by the LUMO energy [78]. The three compounds had similar HOMO energies. Compound **8a** gave the highest HOMO energy, with a slim difference among the investigated compounds. Hence, compound **8a** is anticipated to have the highest tendency to transfer its electrons (Table 5).

The energy gap between the HOMO and LUMO energies represents the chemical stability of compounds. A higher energy gap (∆E) implicates higher chemical stability [82]. In this study, there was a slender difference in the energy gap between the three compounds. Together with this, **8a** had the highest energy gap among the three compounds, but with a slight difference. Hence, compound **8a** is anticipated to have the highest chemical stability among them.

The molecular orbital orientations of the three derivatives were found to be similar to each other. The HOMO orbitals were concentrated mainly on the indole heterocyclic ring. Similarly, the LUMO orbitals were concentrated around the 4-chlorophenyl substituent together with the triazol heterocyclic ring for all the compounds (Figure 8). In general, the HOMO orbitals coincided with the electrophilic reactivity of the MEP regions. By the same token, the LUMO orbitals generally coincided with the nucleophilic reactivity regions, as expected (Figure 8 and Figure 9). The DFT studies have implied that the potential of the compounds for electron transfer might be more susceptible in these regions of the compounds. Hence, these regions might be relatively more prone to the interaction of the compounds with target structures.

## 4. Conclusions

In this article, a series of indole–1,2,4-triazole *S*-linked *N*-arylacetamide structural motifs **8a**–**f** were synthesized in good yields and evaluated for their anticancer potential against the human liver tumor cell line (Hep-G2). The results of the cytotoxic activity evaluation of the synthesized compounds confirmed that anilide ring substitutions increased the anticancer potential of the indole–1,2,4-triazole hybrids. Among the synthesized compounds, the 3,4-dichlorophenyl-based indole–triazole structural hybrid was recognized as the most biologically potent molecule of the series, with the lowest cell viability value (10.99 ± 0.59% at 100 µg/mL) compared with the standard drug ellipticine (cell viability = 11.5 ± 0.55), but displayed comparable potency in comparison with the standard drug doxorubicin (cell viability = 10.8 ± 0.41). The excellent cytotoxicity efficacy against the anti-Hep-G2 cell line was displayed by scaffold **8b**, and the chemotherapeutic efficacy, ADMET, and drug-likeness profile of scaffold **8b** were assessed and confirmed by using molecular docking, MD simulations, and DFT in silico CADD approaches. The in vitro and in silico obtained results and molecular fragment-based SAR studies suggest that this indole-linked triazole hybrid **8b** could be further modified to access more potent molecules via structural modifications. In this regard, compound **8b** is anticipated to possess the highest stability in silico studies and the highest chemotherapeutic potential in vitro anti-Hep-G2 cancer activity, with a promising pharmacokinetic profile. These results suggest indole–1,2,4-triazole **8b** as a promising potential hit lead for the development of an anti-Hep-G2 cancer chemotherapeutic agent.

## Data Availability

All the data of this study are contained in the manuscript and Appendix A.

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
