# Peer review of "Synthesis, Cytotoxic, and Computational Screening of Some Novel Indole–1,2,4-Triazole-Based S-Alkylated N-Aryl Acetamides"

_biomedicines, 2023, doi:10.3390/biomedicines11113078_

Round 1

Reviewer 1 Report

Comments and Suggestions for Authors

The manuscript Synthesis, Cytotoxic and Computational Screening of Some 2 Novel Indole-1,2,4-Triazole based S-Alkylated N-Aryl Acetam-3-ides“ is well written review. Topic is of interest and important for the medicinal chemistry. However following points must be clarified and few modifications are needed.

Minor remarks:

·         line 132: insert space between 0.08 and g

·         line 174: Reference for software MOE is missing

·         lines 263-269: insert space before and after ± as in the Table 1. Write uniformly through the text.

·         Table 1: Explain viability of HepG2 for the compound 8e 123.21 %. What exactly does mean?

·         line 367: In the text (line 350) is indicated that Figure 4 represents molecule 8a in the complex with AKT1, but the title of Figure 4 does not correspond to the above.

·         line 402: Should be indicated that the substances were assessed for the pharmacokinetics and drug-likeness by in silico method.

·         line 406: Explain abbreviations MDCK

·         line 408: Explain meaning of AMES toxic.

·         Table 3: Under the Table 3 should be footnote with explanations of abbreviation TPSA, HIA, MDCK, PAINS. Three decimal places for the value of TPSA are unnecessary. Word Moderate for compound 8f should be put in below row.

·         line 449: Explain abbreviation MEP.

·         Table 5: In title is missing unit for energies.

·         References should be adopted to the Journal style.

Author Response

REVIEWER-1

Response to Reviewer’s Comments

Dear Worthy Editor & Reviewer-1

Biomedicines Journal

Subject:  Synthesis, Cytotoxic and Computational Screening of Some Novel Indole-1,2,4-Triazole based S-Alkylated N-Aryl Acetamides"   (Manuscript ID: biomedicines-2686834)

Dear Sir/Miss

Thank you very much for peer reviewing our manuscript and we appreciate your complimentary recommendations as your comments have helped us significantly to improve the manuscript. We have carefully scrutinized the suggestions mentioned by our worthy reviewer and in accordance of reviewer’s comments, we have revised the manuscript. 

In general, all the recommendations and suggestions have been addressed and incorporated in the manuscript which are yellow highlighted. Following below mentioned are the responses to of reviewer’s comments.

Reviewer Comments

Responses

Point 1. Line 132: insert space between 0.08 and g

Response 1. Dear worthy reviewer, Space inserted between unit g and digit and highlighted yellow in color.

Point 2. Line 174: Reference for software MOE is missing

Response 2. Dear reviewer, the reference has been added in the revised manuscript.

Point 3. Lines 263-269: insert space before and after ± as in the Table 1. Write uniformly through the text.

 Response 3. Dear reviewer, Correction has been made throughout the manuscript as per instruction.

Point 4. Table 1: Explain viability of HepG2 for the compound 8e 123.21 %. What exactly does mean?

Response 4. Dear reviewer, this much cell viability actually reflects that the compound 8e enhances cell viability even more than the control group. This indicates that this compound might be elevating reactive oxygen species (ROS) levels at low concentrations. This, in turn, leads to the idea that other relevant compounds are increasing ROS levels beyond a critical threshold, resulting in cancer cell death. In contrast, this compound appears to increase ROS levels just below that threshold and activates a survival pathway, consequently promoting cell viability. That’s why, it shows

Point 5.   line 367: In the text (line 350) is indicated that Figure 4 represents molecule 8a in the complex with AKT1, but the title of Figure 4 does not correspond to the above.

Response 5. Dear reviewer, the title of the figure has now been corrected.

Point 6. line 402: Should be indicated that the substances were assessed for the pharmacokinetics and drug-likeness by in silico method.

Response 6. Dear reviewer, the suggested change has now been incorporated in the revised manuscript.

Point 7.  line 406: Explain abbreviations MDCK

Response 7. Dear reviewer, MDCK abbreviation is (Madin-Darby Canine Kidney) cells and we have used a computational model of this technique for this study. This information has now been added in the revised manuscript.

Point 8.line 408: Explain meaning of AMES toxic.

Response 8. Dear reviewer, AMES toxicity is a test to assess the potential of compounds and is named after its developer, Dr. Bruce Ames

Point 9.Table 3: Under the Table 3 should be footnote with explanations of abbreviation TPSA, HIA, MDCK, PAINS. Three decimal places for the value of TPSA are unnecessary. Word Moderate for compound 8f should be put in below row.

Response 9. Dear reviewer, the abbreviations of these words and the suggested corrections has been incorporated in the revised MS

Point 10. line 449: Explain abbreviation MEP.

Response 10. Dear reviewer, the abbreviation was explained.

Point 11.Table 5: In title is missing unit for energies.

Response 11. Dear reviewer, the unit for the energy (eV) was incorporated in the revised manuscript.

Point 12.References should be adopted to the Journal style.

Response 12. Dear reviewer, the references adhere to the journal's formatting guidelines.

We hope that revised manuscript would be satisfying for all requirements and will be suitable for consideration for publication.

Kind Regards

Corresponding Authors

Reviewer 2 Report

Comments and Suggestions for Authors

The manuscript presents the synthesis, characterisation and antiproliferative potential of some novel ondole 1,2,4-triazole. Current information regarding the identification of new antitumor compounds in hepatocellular carcinoma is presented.

As comments/suggestions:

1. What can you say about the solubility and stability of the synthesized compounds in solvents compatible with biological environments? Have you carried out such studies?

2. Parallel to the studies carried out on tumor cell lines, studies on normal cell lines should also be carried out to verify the cytotoxicity of the compounds on these types of cells.

3. I suggest the introduction of diagrams or graphs in the manuscript or in the supplementary material for the presentation of the antitumor activity of the compounds.

4. What solvent was used to dissolve the compounds in order to test the antitumor activity?

Author Response

REVIEWER-2

Response to Reviewer’s Comments

Dear Worthy Editor & Reviewer-2

Biomedicines Journal

Subject:  Synthesis, Cytotoxic and Computational Screening of Some Novel Indole-1,2,4-Triazole based S-Alkylated N-Aryl Acetamides"   (Manuscript ID: biomedicines-2686834)

Dear Sir/Miss

Thank you very much for peer reviewing our manuscript and we appreciate your complimentary recommendations as your comments have helped us significantly to improve the manuscript. We have carefully scrutinized the suggestions mentioned by our worthy reviewer and in accordance of reviewer’s comments, we have revised the manuscript. 

In general, all the recommendations and suggestions have been addressed and incorporated in the manuscript which are yellow highlighted. Following below mentioned are the responses to of reviewer’s comments.

Reviewer Comments

Responses

Point 1. What can you say about the solubility and stability of the synthesized compounds in solvents compatible with biological environments? Have you carried out such studies?

Response 1. Dear worthy reviewer, while we have not yet conducted specific studies in this regard, but our designed compounds strongly adhere to the Pfizer Drug Rule, which predicts favorable characteristics for solubility and stability in biological environments, based on these results we can say that they will be compatible in the biological environment.

Point 2. Parallel to the studies carried out on tumor cell lines, studies on normal cell lines should also be carried out to verify the cytotoxicity of the compounds on these types of cells.

Response 2. Dear reviewer, we appreciate your suggestion, and we plan to implement it in our upcoming studies. However, at this point in the manuscript, conducting a new study is not feasible.

Point 3.I suggest the introduction of diagrams or graphs in the manuscript or in the supplementary material for the presentation of the antitumor activity of the compounds.

 Response 3. Dear reviewer, graph is added in manuscript as per your instruction.

Point 4. What solvent was used to dissolve the compounds in order to test the antitumor activity?

Response 4. Dear reviewer, DMSO solvent was used to dissolve the compounds.

We hope that revised manuscript would be satisfying for all requirements and will be suitable for consideration for publication.

Kind Regards

Corresponding Authors
